# Long-Term Effects of Kasai Portoenterostomy for Biliary Atresia Treatment in Russia

**DOI:** 10.3390/diagnostics10090686

**Published:** 2020-09-11

**Authors:** Anna Degtyareva, Alexander Razumovskiy, Nadezhda Kulikova, Sergey Ratnikov, Elena Filippova, Ekaterina Gordeeva, Marina Albegova, Denis Rebrikov, Anna Puchkova

**Affiliations:** 1Kulakov National Medical Research Center for Obstetrics, Gynecology and Perinatology, 4 Oparina Street, 117997 Moscow, Russia; annadim@yahoo.com (A.D.); filippova_ea@oparina4.ru (E.F.); albegova_mb@oparina4.ru (M.A.); puchkova_aa@oparina4.ru (A.P.); 2Neonatology Department, Sechenov University, 8 Trubetskaya Street, 119991 Moscow, Russia; 3Center for Precision Genome Editing and Genetic Technologies for Biomedicine, Pirogov Russian National Research Medical University, 1 Ostrovityanova Street, 117997 Moscow, Russia; 1595105@mail.ru; 4Filatov Munitsipal Children’s Hospital, 15 Sadovaya-Kudrinskaya Street, 123001 Moscow, Russia; kulikova_nv@filatovskaya.ru (N.K.); gordeeva_ea@filatovskaya.ru (E.G.); 5National Medical Research Center for Children’s Health, 2 Lomonosovsky Prospect, 119296 Moscow, Russia; ratnikov_sa@nczd.ru

**Keywords:** biliary atresia, Kasai portoenterostomy, cholangitis, portal hypertension, bile ducts dilatations, native liver survival

## Abstract

This prospective study enrolled 144 patients after surgical treatment of biliary atresia in early infancy. We analyzed the immediate effectiveness of the surgery and the age-related structure of complications in the up to 16-year follow-up. The immediate 2-year survival rate after the surgery constituted 49.5%. At the time of this writing, 17 of the patients had celebrated their 10th birthdays with good quality of life and no indications for transplantation of the liver. The obtained results underscore the critical importance of surgical correction of biliary atresia by Kasai surgery in the first 60 days of life and subsequent dynamic follow-up of patients for the purpose of the early detection and timely correction of possible complications.

## 1. Introduction

Till the mid-20th century, biliary atresia (BA) was lethal [1,2]. Surgical correction of this defect by portoenterostomy was introduced in 1952 by Prof. Morio Kasai. The operation includes resection of fibrotic lesions in the portal area and re-establishment of the physical connection of the liver with the intestine by a Roux loop anastomosis, resulting in restoration of bile flow to the intestine [3]. This operation is considered as a maintenance treatment for the majority of BA cases. It prolongs the native liver survival in preparation for liver transplantation (LT). Current estimations of 5-year survival rates for BA patients with native liver are 30–70% [4,5,6,7]. In most cases, Kasai surgery is a palliative treatment for children with BA that prolongs life with the native liver and is an important step in preparing for liver transplantation [4]. Notably, in some patients, the Kasai procedure alone can provide a good functional state of the liver for more than 20 years, and the longest to date follow-up of a native liver survivor with minimal liver symptoms is over 60 years [6]. At the same time, direct connection of intrahepatic bile ducts with the intestine greatly increases the risks of cholangitis, which is observed in 45–87% of the patients [4]. Other complications arising in the follow-up include portal hypertension, biliary cysts, and hepatopulmonary syndrome [1,4,7].

BA is the most common indication for LT in children [6]. According to a multi-center study involving 1911 patients at 39 clinics in North America in 2011–2018, BA patients constitute 38.5% of liver transplant recipients under the age of 18 [8].

The study is aimed at evaluation of functional condition of the liver, rates of survival as a function of the age at surgery, and the age-related structure of complications in the follow-up of Kasai surgery.

## 2. Materials and Methods

One hundred and forty-four infants (61 boys, 42.4%, and 83 girls, 7.6%), weighing 3.24 ± 0.53 kg at birth [min 1.50; max 4.25], participated in the prospective study; 13 of the patients were born prematurely (at 34–36 wGA). The patients were diagnosed with BA on the basis of standard complex examinations, including morphological examination of the liver and biliary system biopsies. Biopsy specimen histology was performed according to the standard Van Gieson staining protocol. In some cases, immunohistochemistry (cytokeratins 7 and 19 monoclonal antibody) was performed to clarify the diagnosis. The liver parenchyma inflammatory process was controlled by histological activity index (Knodell scale). The fibrosis stage was calculated by Desmet histological index. Hematological studies were carried out at Sysmex XT 800i and 4000i (Sysmex, Japan) by the method of fluorescence flow cytometry. Biochemical studies were carried out in serum by spectrophotometric and turbidimetric methods at BA-400 (Biosystems, Barcelona, Spain) using original reagents.

All patients enrolled in the study manifested signs of bile duct atresia, rudimentary gallbladder and characteristic hepatic alterations in the form of cholestasis, intrahepatic bile duct proliferation, signs of fibrosis, and portal and periportal inflammation. Intraoperative cholangiography (when indicated) revealed the signs of bile duct obstruction and the lack of a patent extrahepatic bile duct. In addition, all candidates were examined for TORCH infections and congenital disorders manifested by neonatal cholestasis, which could be comorbid or intercurrent with BA. The tests encompassed blood levels of α1-antitrypsin, urinary bile acid profiles, plasma amino acid and acyl carnitine profiles, urinary succinyl acetone, urinary and plasma oxysterols, and plasma galactose and galactose-1-phosphate uridyl transferase activity levels. Alagille syndrome was excluded by echocardiography, ophthalmological examinations, and X-rays of the spine. Patients with neonatal cholestasis of different etiology, patients who did not undergo surgery, as well as those with incomplete follow up, were excluded from participation.

The patients underwent Kasai surgery in 2000–2020 at the age of 79.4 ± 21.5 days of life [min 27; max 138]. The in-patient treatment started under the age of 3 months for all patients. Upon admission, all patients manifested jaundice, colorless stools, and increased hepatic and splenic volumes.

The open Kasai procedure with anti-reflux was performed for 34 (23.6%) of the patients. The reconstruction of the anti-reflux valve involved circular deserosation of a 1 cm length of the intestine at the junction with the Roux loop and intussusception with sero-serosal nodal sutures. In addition, 63 (43.8%) of the patients were operated laparoscopically and 47 (32.6%) were operated through a mini-laparotomy incision. During the Kasai surgery, by mini-access (mini-laparotomy), the surgical field is accessed via a 2–3 cm incision in the right hypochondrium. Compared with open surgery, this approach does not require mobilization of the liver (by intersection of the sickle ligament); the rest of the surgical steps are identical. The mini-laparotomy approach combines the advantages of open surgery (the possibility of using higher magnification, enhanced hermeticity of the anastomosis, and reduced operation times) and laparoscopic techniques (low invasiveness, minimal adhesions in the abdominal cavity, and excellent cosmetic results) [7].

In the postoperative period, all patients received multimodal analgesia, infusion therapy with partial parenteral nutrition, ursodeoxycholic acid, antibacterial and symptomatic therapies, and glucocorticoid therapy in accordance with established guidelines [9]. In order to prevent cholangitis post-operatively, prophylaxis with trimethoprim-sulfamethoxazole was administered on a long-term basis.

The postoperative in-patient care lasted for 24.8 ± 11 days [min 9; max 69].

The immediate effectiveness of the surgery was determined by coloration of stools, jaundice reduction, and decreased bilirubin levels.

The follow-up encompassed:Evaluation of functional condition of the liver (cholestatic syndrome manifestations, increased transaminase levels, and indicators of synthetic activity of the liver) with relation to age;Analysis of immediate or delayed complications (postoperative complications, bacterial cholangitis, bile ducts dilatation, portal hypertension, hepatopulmonary syndrome) with relation to age;Survival study.

Cholangitis was diagnosed on the basis of febrile fever symptoms, elevated serum levels for the markers of systemic inflammation (C-reactive protein, procalcitonin), alterations in ESR, WBC counts and WBC differential, in combination with varying degree of clinical and biochemical manifestations of cholestatic syndrome, increased transaminases, and reduced synthetic function of the liver.

Recurrent cholangitides that did not respond to conservative therapies were treated by surgical reconstruction of the anti-reflux valve at the Roux loop-intestine junction. The procedure aimed at reducing contamination of the portal area with intestinal flora. It involved circular deserosation of a 1 cm length of the intestine and reconstruction of the valve by intussusception with sero-serosal nodal sutures.

Portal hypertension was assessed on the basis of impaired blood flow in the portal system (umbilical vein recanalization, ascites, enlarged spleen, and the altered volume rate of portal blood flow) as revealed by Doppler ultrasound examination, esophageal varices, and hematological indicators of hypersplenism (thrombocytopenia, anemia, leukopenia). Endoscopic examination was performed routinely once in 1–1.5 years or upon appearance/aggravation of hematological (thrombocytopenia and/or pancytopenia) and sonographic signs (enlarged spleen, recanalized umbilical vein, decreased rates of portal blood flow, increased rates of splenic venous blood flow, splenomegaly, ascites) of portal hypertension.

Intrahepatic bile duct (iHBD) dilatations were assessed by ultrasound examination.

The follow-up duration varied from 6 months to 16 years.

The data were processed in StatSoftStatistica 10 (StatSoft, Inc., Tulsa, OK, USA) and Microsoft Excel 2016 software. Numerical variables were described by mean and standard deviation values (mean ± SD). Categorical variables were described by absolute numbers and frequencies of the events. Univariate comparisons for two dependent groups were made by non-parametric Wilcoxon test. Bilateral Fisher’s exact test was applied for the comparison of frequencies between the groups. The survival was described by Kaplan-Meier curves. The differences were considered statistically significant at *p* < 0.05.

The study protocol was reviewed and approved by the Local Ethics Committee of the Pirogov Russian State Medical University (Protocol No.2002/18 from Sept 02 2002); the study was conducted in accordance with the Declaration of Helsinki. All participants (children’s parents) provided written informed consent.

## 3. Results

### 3.1. Restoration of Liver Function

The age of the patients at Kasai surgery constituted 79.4 ± 21.5 days. The majority of patients (77, corresponding to 53%) were operated upon at the age of 61–90 days. Of the rest, 32 infants (22%) were operated at a younger age (<60 days of life) and 36 infants (25%) were operated at > 90 days of life.

In postoperative in-patient care, serious complications were developed by 7 patients (4.9%). One patient had duodenal perforation on day 4 post-operation (p/o), and two patients had colon perforations on days 3 and 7 p/o; these patients were re-operated upon. One patient had adhesive intestinal obstruction treated surgically on day 15 p/o. Gastrointestinal bleedings in three other patients were treated conservatively. One patient died of colon perforation on day 12 p/o.

The establishment of a physical connection between the biliary tree and the intestine, indicated by coloration of stools, was observed in 128 patients (89.5%). Blood test results for these patients before the surgery and on day 10 p/o are given in Table 1. The coloration of stools typically occurred on day 3–4 p/o; in a few cases, it occurred on day 1 p/o. In 2 patients, the coloration of stools occurred on days 30 and 34 p/o.

The effectiveness of surgical intervention (as assessed by coloration of stools combined with jaundice disappearance or reduction with a concomitant decrease in bilirubin levels in the early post-operative period) constituted 75% (108 patients). During post-operative in-patient care, jaundice was alleviated in 90 patients and totally suppressed in 28 patients.

For 36 patients (25%), the operation was ineffective, and the situation eventually led to biliary cirrhosis with lethal outcomes for 2 patients (5.6%) at the ages of 5 and 7 months. The other 34 patients with ineffective Kasai surgery received liver transplants at the age of 14.6 ± 2.5 months.

By the end of the first year of life, the positive effect was maintained in 98 patients. The remaining 10 patients with the pronounced positive effect in the early post-operative period eventually developed colorless stools and signs of liver cirrhosis. The age-related dynamics of liver function indicators evaluated with the exclusion of liver transplant recipients are given in Table 2.

By the age of 1 year, total bilirubin was reduced to normal levels in 25 patients with effective Kasai surgery (constituting 11.8 ± 2.3 µmol/L). In 73 patients, bilirubin was still elevated (98.4 ± 33.6 µmol/L) but significantly and stably reduced, as compared with the initial levels before the surgery. By the age of 2–3 years, bilirubin levels were reduced to normal values in all patients with effective Kasai procedure (Table 2).

By the age of 1 year, GGT activity was reduced to normal values in 22 patients. In 76 patients, it was still elevated but reduced significantly, as compared with the initial level. By the age of 3 years, 86% of the patients had normal GGT levels. Several cases of persistence of elevated GGT levels should be noted (up to 412.3 U/mL, Table 2), although such patients had normal levels of bilirubin and other cholestasis markers, and showed no indications for LT.

By the age of 1 year, the activity of ALT was reduced to a normal value (17 U/L) only in 1 patient. For the rest, it constituted 109.12 ± 84.65 U/L. The activity of AST was elevated in all 1-year old patients (105.17 ± 59.2 U/L, Table 2). By the age of 4 years, 65% of the patients had normal levels of transaminase activity. However, the moderate elevation of ALT/AST activity persisted in 14% of the patients for at least 3–5 years after the surgery.

Thus, the majority of BA patients with effective Kasai surgery showed normal bilirubin levels by the age of 1 year, while the reduction in the activity of GGT, ALT, and AST took much longer, and the elevated blood plasma levels for these enzymes typically persisted (for >5 years in some patients).

### 3.2. Complications

The age-related structure of complications is shown in Figure 1.

Bacterial cholangitides and portal hypertension occurred most typically. Cystic dilatations of intrahepatic bile ducts were less common. One patient developed hepatopulmonary syndrome, which is seen as an indication for LT, at the age of 3 years.

### 3.3. Cholangitides

In the 1st year after surgery, 102 of 143 patients (71%), regardless of the effectiveness of the surgical intervention, had at least one episode of cholangitis which required in-patient treatment with intravenous administration of antibacterials (Figure 1). It is important to note that all patients with cholangitis developing against the backdrop of ineffective Kasai surgery manifested coloration of stools in the early post-operative period. The establishment of physical connection between the liver and the intestine, whatever transient, inevitably creates the risk of contamination of bile ducts with intestinal flora. In patients with cholangitis against the backdrop of ineffective Kasai surgery, the appearance of stained stools usually persisted for several weeks, but was not accompanied by jaundice alleviation. Dilatations of bile ducts, revealed more commonly in the patients with ineffective surgery (5 cases out of 35, i.e., 14% vs. 6% for the total sample, Figure 1), could exacerbate the inflammatory reaction by promoting bacterial growth. Seven patients with recurrent cholangitides were re-operated for antireflux valve insertion, which was effective. In 19 patients with effective Kasai surgery, the recurrent cholangitides promoted the onset of liver cirrhosis, which served as an indication for LT performed at the age of 9–14 months.

In the 2nd year of life, 54 of 98 patients (55%) had episode(s) of cholangitis requiring the in-patient antibacterial therapy (Figure 1). Two of the patients underwent the anti-reflux valve surgery at the age of 18 and 19 months. In 24 patients, cholangitis led to biliary cirrhosis; these patients received liver transplants at the age of 18–28 months.

During the 3rd year of life, acute cholangitis was diagnosed in 20 patients (33%, Figure 1). Three of those developed weak signs of systemic inflammatory reaction, but acholic stools were observed in none of the cases. The episodes were successfully resolved by oral administration of antibacterials on an out-patient basis.

During the 4th and 5th years of life, cholangitis was diagnosed in 11 patients (22%, Figure 1). For the age groups of 5–10 and > 10 years, the occurrence constituted 15% and 16%, respectively. In the latter group, only 2 patients had single episodes of hepatocellular dysfunction with portal hypertension at the age of 12 and 14 years (despite the ongoing observation and maintenance therapy), which served as indications for LT.

### 3.4. Portal Hypertension

During the 1st year after surgery, 67 of 143 patients (47%) were diagnosed with portal hypertension (PH, Figure 1). Of those, 26 patients (38.8%) showed minimal signs of umbilical vein recanalization and 20 patients (29.9%) developed ascites. It should be noted that recanalization of umbilical vein frequently clears on its own during infancy—in 24 patients, it was transient and cleared by the age of 1.5 years. Esophageal varices (EV) were identified in 22 patients (32.8% of PH cases, including 10 cases of EV grade I, 8 cases of EV grade II, 3 cases of EV grade III, and 1 case of EV grade IV). Seven patients underwent EV sclerotherapy and 6 patients had EV bleeding episodes during the first year. Hypersplenism with thrombocytopenia was identified in 7 cases (10.4%), one of them also with the signs of anemia. One patient with well-preserved liver function developed a therapy-resistant ascites and received LT at the age of 11 months. One patient had gastrointestinal bleeding with lethal outcome at the age of 7.5 months.

During the 2nd year of life, 43 patients (44%, Figure 1) manifested signs of PH, including recanalized umbilical vein (rUV, 5 cases, 11.6% of PH cases), ascites (4 cases, 9.3%), and/or EV (14 cases, 32.6%, including 2 cases of EV grade I, 7 cases of EV grade II, and 1 case of EV grade III). Three patients had EV bleedings and received sclerotherapy. Hypersplenism was observed in 18 cases (41.9% of PH cases), with related splenic thrombocytopenia in 14 cases and a combination of thrombocytopenia and anemia in 5 cases.

Two patients, 1.5 and 2.5 years old, diagnosed with EV grade III–IV with high risks of bleeding, received a small-diameter splenorenal shunt, which ensured partial drainage of the blood from portal circulation into the vena cava inferior. This intervention mitigated the severity of EV to grade I–II.

During the 3rd year of life, 24 patients (39%, Figure 1) manifested signs of PH including rUV (5 cases, 20.8% of PH cases), ascites (2 cases, 8.3%), and/or EV grade I–III (15 cases, 62.5%, including 7 cases of EV grade I, 6 cases of EV grade II, and 2 cases of EV grade III). Three EV patients received sclerotherapy. Two patients had esophageal bleedings, the conservative treatment of which was successful. The signs of hypersplenism were observed in 7 cases (29.2%), related splenic thrombocytopenia was identified in 8 cases (33.3%), and splenic thrombocytopenia combined with anemia was identified in 2 cases of PH (8.3%).

During the 4th and 5th years of life, 20 patients (44%, Figure 1) manifested signs of PH, including rUV (11 cases, 55% of PH cases), EV (16 cases, 80%, including 8 cases of EV grade I, 5 cases of EV grade II, and 3 cases of EV grade III), and/or thrombocytopenia (9 cases, 45%).

Among the 5–10 year olds, 22 patients (70%, Figure 1) manifested signs of PH, including both EV (22 cases, 100% of PH cases, including 8 cases of EV grade I, 8 cases of EV grade II, and 4 cases of EV grade III) and rUV (22 cases, 100%). Three of the patients had EV bleeding, and one patient received sclerotherapy. The majority of patients also had hypersplenism with thrombocytopenia (20 cases, 91%).

At the age of > 10 years, 13 of 17 patients (76%, Figure 1) manifested signs of PH including rUV in 12 cases (92.3% of PH cases), EV in 13 cases (100%, including EV grade I in 5 cases, EV grade II in 5 cases, and EV grade III in 3 cases), hypersplenism with thrombocytopenia in 13 cases (100%), and hypersplenism with thrombocytopenia combined with anemia in 10 cases (76.9%). Two of the patients had EV bleedings, and one patient received sclerotherapy.

### 3.5. Intrahepatic Bile Duct Dilatations

iHBD dilatations were revealed in 9 patients in the 1st year after surgery (6.2%), in 10 patients in the 2nd year of life (12.2%), and in 6 patients in the 3rd year of life (9.8%, Figure 1). The majority of cases represented moderate dilatations (0.5–4.5 mm) without signs of biliary obstruction. Three patients developed 30–40 mm iHBD dilatations against a background of recurrent cholangitis; these patients were re-operated to apply cystoenteroanastomoses on the basis of the existing Roux loop. The interventions were successful and prevented further growth of the iHBD cysts, as revealed by dynamic ultrasound examinations.

In patients aged >3 years, moderate iHBD dilatations (1.5–2 mm) were observed with no signs of biliary obstruction or any other indications for the surgery (Figure 1). The dilatations were revealed in 8 patients aged 3–5 years (15.7%), 6 patients aged 5–10 years (19.4%), and 3 patients aged >10 years (17.6%). It should be noted that, in contrast to biliary cirrhosis, none of the observed iHBD dilatations interfered with synthetic function of the liver, and this condition can be generally interpreted as a consequence of partial biliary obstruction prior to the portoenterosromy and in association with it.

### 3.6. Survival Study

Of all the participants, 89.5% survived till the age of 5 months. In the native liver (non-LT) subgroup, survival rates for 1-, 2-, 3-, 5-, 10-, and over 13-year period constituted, respectively, 72.9, 49.5, 45.5, 40.6, 34.6, and 28.7% (Figure 2).

For the native liver (non-LT) subgroup stratified by age at Kasai surgery, survival rates for the patients operated at the age of under 60 days were higher. Of those differences, however, only the difference in 2-year survival rates was significant (*p* < 0.05, Figure 3). It should be noted that survival rates for the <1 year-old showed no correlation with age at Kasai surgery (Figure 3).

Advanced stratification of the sample by age at Kasai surgery (Figure 4) revealed a similar pattern. By years 2 and 3 after the surgery, survival among the patients operated at the age of under 60 days was significantly higher as compared with patients operated at 60–90 days of life (*p* = 0.03 and *p* = 0.04, respectively, Figure 4), and no significant differences were observed for the other age groups.

## 4. Discussion

Kasai surgery is the gold standard of palliative therapy for BA as it efficiently allows to circumvent the vital need for LT in young infants born with BA. At the same time, Kasai surgery gives a chance of postponing transplantation indefinitely.

Transplantation of the liver (as of any vital organ or its part) is never performed as a preventive measure; the indications for LT include significant risks of lethal outcome and inefficiency of alternative treatments. LT involves complex surgery with the risk of complications in the intra- and post-operative periods. The recipients remain under close observation for the rest of their lives and, in addition, must receive life-long immunosuppression therapies. LT is not just a ‘do-and-forget’ operation, and the quality of life with native liver is always better.

Before 2008, LT was not performed on young children in Russia and the only possibility was to receive it abroad. Noteworthy, the state health insurance program covered such operations; however, the paperwork for the operation abroad took a critically long time in many cases. The situation has cardinally improved with the establishment of LT protocols for young children. The operation typically involves LT from living donors (relatives) and is carried out in a timely manner upon the indications.

The current BA patient management standards in Russia are similar to those in other countries. In the case of ineffective Kasai surgery, the patient is subject to intensive treatment in preparation for LT, the optimal timing of which is determined individually based on a sum total of medical factors. Patients with effective Kasai surgery receive supportive therapies while remaining under dynamic supervision aimed at the early detection of complications and their timely correction. If the condition of a patient worsens with the appearance of clear indications for LT, the transplantation is performed on an urgent basis, regardless of the age and the fact of Kasai surgery in the anamnesis; the expenses are covered by the federal health insurance program, i.e. the operation is free for the patient.

In this study, we assessed the effectiveness of Kasai surgery by post-operative resolution of BA symptoms, age-related complications, and survival rates. According to the published evidence, the effectiveness of Kasai surgery generally varies in the range of 50–95% [10,11,12,13,14,15,16]. In some studies, the effectiveness is assessed by a decrease in bilirubin levels to 2 mg/dL (34.2 µmol/L), which renders the effectiveness of 40% on average [15].

In our experience, Kasai surgery caused coloration of the stools within 4–8 days p/o. Alleviation of jaundice and reduction in bilirubin levels progressed gradually and took much longer. The increased activity of GGT and transaminases in the early postoperative period was observed in the majority of patients with effective Kasai surgery (as compared to the values before the surgery). In our opinion, this may reflect the compensatory reaction of the body to the surgical intervention per se, as well as to the use of potentially hepatotoxic pharmaceuticals (anesthetics, antibacterials, etc.). In the follow-up, the GGT/ALT/AST activities gradually decreased. Their long-term dynamics may be relevant. As demonstrated by Ihn et al., high plasma levels of GGT (above 500 U/L, persisting for at least 5 months and accompanied by jaundice) represent a poor prognostic factor, which significantly shortens the native liver life expectancy [17]. At the same time, Noor et al. revealed no correlation of total bilirubin levels and GGT/ALT activities in the postoperative period with native liver life expectancy [18]. According to our experience, in 86% of the patients with effective Kasai surgery, GGT levels decreased slowly and reached the reference range only by the age of 3 years. At the same time, in 89% of the patients, the majority of them having elevated GGT levels, bilirubin decreased to normal levels by the end of the first year. In the remaining 11% of the patients, bilirubin decreased to normal levels by the age of 2–3 years. In a longer follow-up, none of the >4 year-olds had bilirubin levels >58 µmol/L even during episodes of cholangitis. In contrast to bilirubin, transaminase levels were reduced to normal values in only 65% of the 4 year-old patients, while 14% of the patients showed moderate elevation of transaminase levels for up to 5 years (and in few cases even longer). It should be noted that in such cases transaminase levels show no age-related dynamics and apparently have no prognostic significance.

The observed correlation of the native liver survival rates with the age at surgery is consistent with the literary data on this subject [11,13,16,19]. For instance, Wang et al. demonstrated that Kasai surgery at the age of under 81 days significantly increases native liver life expectancy as compared with the patients operated at a later age [13]. In a large-scale multicenter study carried out in France with the enrollment of 1044 patients during the 1986–2009 period, the 5-, 10-, and 20-year native liver survival rates constituted 40%, 36%, and 30%, respectively [16]. In a study by Liu et al. (2017), the survival constituted 84% and 71% for 1 year and 2 years of life, respectively [19]. Nio et al. (2012) reported a 20-year survival of 33% [20]. Some published observations of native liver survival encompass 26 years [21] and 40 years [6], and most remarkably over 60 years [6,21] after Kasai surgery.

The predominant complications in the follow-up were recurrent cholangitides and portal hypertension. Of the total sample of operated BA patients (independently of the surgery effectiveness), 71% suffered at least one episode of cholangitis in the first year after surgery. The observed age-related decrease in the frequency and severity of cholangitides is consistent with the literary data. As shown by Lee et al., 51.5% of the episodes arise during the initial 180 days p/o, followed by 21.6% between days 180–365 p/o, 11.3% between months 12 and 18 p/o, and 15.5% between months 18 and 24 p/o [22]. Despite the diversity of possible mechanisms, the most common cause of cholangitis in the follow-up of Kasai surgery is the ascending infection from the small intestine via the portoenteroanastomosis [22,23]. It is believed that the expanding bacterial flora suppresses the albumin synthesis and promotes the accumulation of ammonia, which additionally facilitates bacterial expansion. Cystic dilatations of intrahepatic bile ducts are also favorable for the inflammation. Besides, retention of bile in the dilated ducts and small cysts favors the formation of microconcrements, which may damage the biliary tree and trigger the dormant inflammatory reaction. High intraluminal pressure of the small intestine may also contribute to cholestasis, growth of pathogenic microflora, and, ultimately, ascending cholangitis [24]. Plausible contribution of direct surgical side-effects e.g. ischemic lesions in the stitching area and inflammatory response from the biliary tract [25] should be mentioned as well, especially since we observed cholangitides in the early postoperative period not only in the cases of effective Kasai surgery, but also in the cases of ineffective surgery when the persistent paleness of stools in the postoperative period indicated the lack of communication between the biliary system and the intestine. As reported by Selvalingam et al., the probability of cholangitides in the early postoperative period also depends on the diameter of bile ducts within the operative field: smaller diameters (<150 µm) correlate with higher frequencies of cholangitides [26].

Portal hypertension (PH) of diverse pathophysiology (from the moderate splenomegaly and umbilical vein recanalization, which minimally affected life quality, to the ascites and severe esophageal varices) was manifested by 47% of the patients in their first year after the surgery. Later on, the occurrence of PH increased, reaching 76% after the age of 10 years. The primary cause of PH is hepatic fibrosis, which develops before the operation and subsequently becomes aggravated by recurrent cholangitides. According to the reported evidence, portal hypertension in the follow-up of Kasai surgery occurs in 37–70% of the patients [27,28]. Importantly, for patients with effective Kasai surgery, even the clinically evident PH was not considered as an indication for emergency liver transplantation, as the varices were subject to scheduled surgical correction, thrombocytopenia was not accompanied by hemorrhages, and no indications for blood transfusion were encountered. Apparently, PH is a primary consequence of the fibrotic alteration of hepatic tissues within the portal area. Therefore, PH episodes are inherent for BA, and can only be mitigated (not completely resolved) by successful Kasai surgery. Strictly speaking, PH should be considered as an echo of deteriorating changes that would rapidly progress in the absence of timely surgical intervention (and this point is consistent with our observations for the cases of ineffective surgery). However, a detailed discussion of this issue is beyond the scope of this study.

## 5. Conclusions

The results of this study support the view of Kasai surgery as a crucial maintenance measure for BA patients, which is also the essential precondition of native liver survival. The immediate 2-year survival rate after the surgery constituted 49.5%. At the time of this writing, 17 of the patients have celebrated their 10th birthdays with good quality of life and no indications for LT. For less successful outcomes, the Kasai procedure still allows to delay LT for at least few months, considering that LT effectiveness largely depends on the age and satisfactory physiological condition of the patient.

The most common complications in the follow-up are recurrent cholangitides ultimately leading to liver cirrhosis. The risks of developing cholangitides are high during the first three years of life and decrease with the patient’s age. The risks of portal hypertension, by contrast, increase with the patient’s age; however, in the majority of cases, this complication has no critical influence on the functional condition of the liver, and it is not an indication for LT.

The obtained results underscore the critical importance of surgical correction of BA by Kasai surgery during the first 60 days of life and subsequent dynamic follow-up of the patients for the purpose of the early detection and timely correction of possible complications.

## Figures and Tables

**Figure 1 diagnostics-10-00686-f001:**
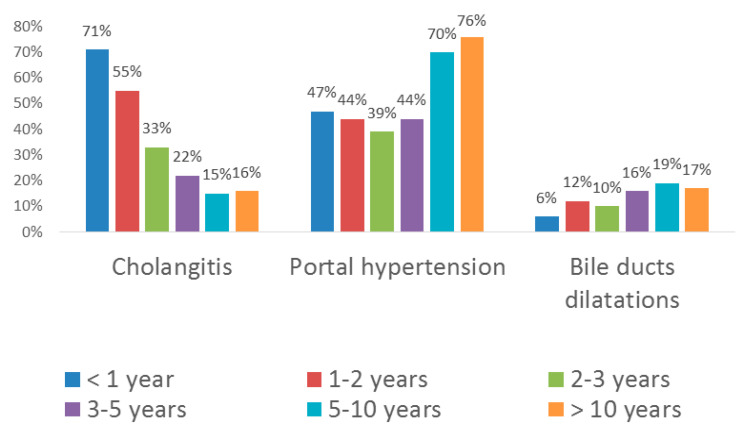
Age-related structure of complications observed in BA patients after Kasai surgery (percent values reflect the proportion of patients affected by the condition, not the number of episodes).

**Figure 2 diagnostics-10-00686-f002:**
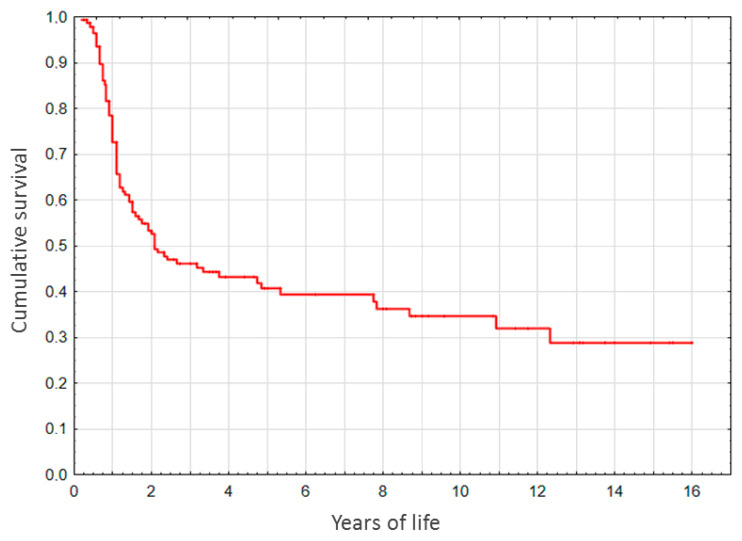
Survival with native liver (non-LT subgroup) after Kasai surgery.

**Figure 3 diagnostics-10-00686-f003:**
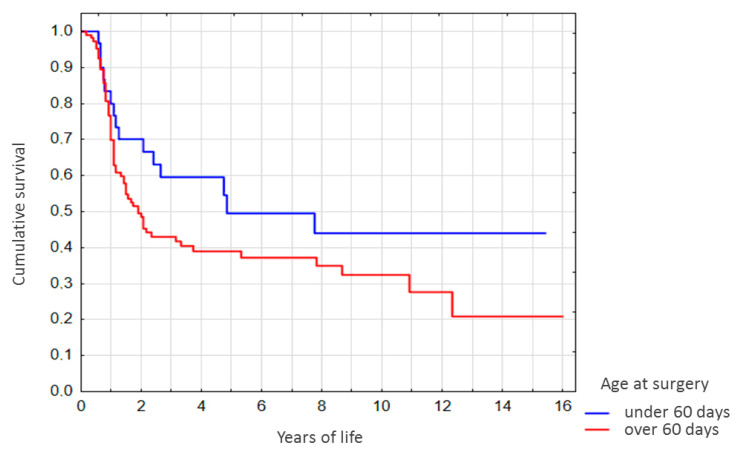
Survival curves for the non-LT subgroup stratified by age at Kasai surgery.

**Figure 4 diagnostics-10-00686-f004:**
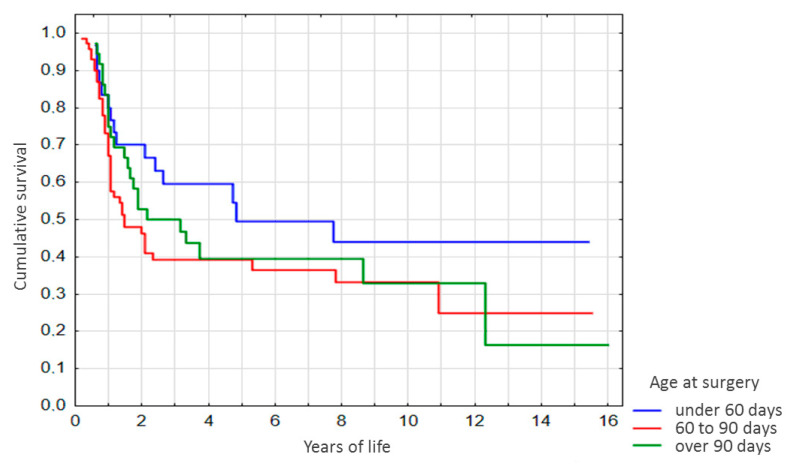
Survival curves for the non-LT subgroup with advanced stratification by age at Kasai surgery.

**Table 1 diagnostics-10-00686-t001:** Blood test indicators before the surgery and on day 10 p/o for patients with coloration of stools in the early post-operative period (*n* = 128).

Indicators	Before Surgery $\bar{\chi }\pm \;s\;$ [Min; Max]	After Surgery $\bar{\chi }\pm \;s\;$ [Min; Max]	Wilcoxon Test Results
GGT, U/L	634 ± 347 [398; 1805]	989 ± 496 [96; 2134]	*p* < 0.05
ALT, U/L	137 ± 94 [70; 286]	266 ± 179 [81; 663]	*p* > 0.05
AST, U/L	236 ± 99 [142; 339]	253 ± 118 [74; 447]	*p* > 0.05
Bilirubin total, µmol/L	211 ± 67 [123; 418]	93 ± 58 [5; 342]	*p* < 0.05
Bilirubin direct, µmol/L	108 ± 46 [77; 244]	58 ± 42 [1; 202]	*p* < 0.05
Cholesterol, mmol/L	5.5 ± 1.6 [1.4; 11.6]	5.5 ± 2.2 [1.7; 15.1]	*p* > 0.05
Fibrinogen, g/L	2.5 ± 0.6 [1.3; 4.2]	2.3 ± 0.7 [1.3; 4.7]	*p* > 0.05
PI, %	90 ± 12 [77; 120]	96 ± 18 [50; 140]	*p* < 0.05
Albumin, g/L	38 ± 5 [28; 47]	37 ± 5 [27; 51]	*p* > 0.05
Cholinesterase, U/L	6262 ± 1989 [2509; 11776]	5123 ± 1634 [2391; 11129]	*p* > 0.05

**Table 2 diagnostics-10-00686-t002:** Age-dependent dynamics of native liver function indicators in the follow-up of Kasai surgery (the values are given as $\bar{\chi }$ ± s [min; max]).

Age, Years	n	GGT, U/L, $\bar{\chi }\pm \;s$ [Min; Max]	ALT, U/L, $\bar{\chi }\pm \;s$ [Min; Max]	AST, U/L, $\bar{\chi }\pm \;s$ [Min; Max]	Bilirubin Total, µmol/L, $\bar{\chi }\pm \;s$ [Min; Max]	Bilirubin Direct, µmol/L, $\bar{\chi }\pm \;s$ [Min; Max]
1	98	148 ± 95 [36; 327]	136 ± 99 [16; 381]	107 ± 62 [39; 376]	58 ± 99 [9; 381]	33 ± 65 [1; 251]
2	61	166 ± 175 [8; 1275]	131 ± 101 [15; 371]	111 ± 66 [39; 377]	61 ± 114 [6; 401]	35 ± 49 [1; 229]
3	51	121 ± 88 [8; 288]	82 ± 68 [18; 239]	94 ± 57 [33; 198]	20 ± 11 [7; 45]	9 ± 12 [2; 39]
5	31	93 ± 99 [9; 388]	45 ± 43 [17; 152]	51 ± 36 [4; 144]	14 ± 7 [5; 31]	4 ± 2 [1; 11]
10	19	139 ± 120 [15; 398]	46 ± 36 [20; 97]	55 ± 31 [22; 233]	14 ± 8 [6; 38]	5 ± 3 [2; 13]
>10	17	74 ± 86 [16; 147]	22 ± 4 [19; 25]	35 ± 6 [30; 45]	19 ± 6 [14; 28]	8 ± 1 [3; 10]

## Data Availability

The datasets used and/or analyzed during the current study are available from the corresponding author on reasonable request.

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
