# Peer review of "Long-Term Effects of Kasai Portoenterostomy for Biliary Atresia Treatment in Russia"

_diagnostics, 2020, doi:10.3390/diagnostics10090686_

Round 1
Reviewer 1 Report
I read with interest the paper from Degtyareva et al. The caseload of study, about 7 pts/years, is good to built a solid experience with biliary atresia patients
Nevertheless this paper presents several conceptual errors and misinterpretations of the data . Just a few examples:
- part of the results have been presented in the section of Materials and Methods
- the section of Materials and Methods is lacking of definitions of some issues presented in the Results section.
- Intrahepatic biliary dilatations are not equivalent to to intrahepatic biliary lakes.
- which is the difference between open Kasai and minilaparotomy incision? which type of antireflux valve was used and in all patients or only in patients with open surgery?
- Transaminases are not good biological marker of cholestasis
- some results are not reliable : nobody all around the world has 89.5% of good post-operative bile drainage after kasai procedure. Color of stool and "decreased bilirubin level" are somewhat elusive and subjective parameters. A certain degree of stool staining surgery should not be mistaken for good post-operative bile drainage. To confirm this , Authors report that only 28 patients (19.4%) have become jaundice free in the post-operative; that is not a good result.
- Authors found no difference in the incidence of cholangitis in patients with and without good post-oeprative bile drainage. This is in conflict with the experience reported in the literature by all the Authors according to whom cholangitis occurs almost exclusively in patients with a good biliary drainage. And beside, although cholangitis itself is one of the main causes of the cessation of biliary drainage and failure of the Kasai procedure , Authors do not mention , how they try to prevent its oneset
- The data presented in the tables are not clear
- since one of the still most debated issues is the use of laparoscopic technique compared to open technique, in the analysis of the results Authors do not analyze the results of the different techniques
- about portal hypertension:do the authors perform routine endoscopic evaluation? If no, which are the indications for scoping?
- Discussion:In view of the above, the discussion should also be reviewed
- there are typing errors even in the literature
Author Response
Dear Reviewer,
Thank you for the thorough consideration and valuable comments on the manuscript.
The Review raises several important issues which we tried to resolve in the revision.
Here is a list of what has been done:
- Comment: the section of Materials and Methods is lacking definitions of some issues presented in the Results section.
Response: The Materials and Methods section was expanded.
- Comment: what is the difference between open Kasai and minilaparotomy incision?
Response: In the Kasai surgery 'from mini-access' (mini-laparotomy), the surgical field is accessed via a 2-3 cm incision in the right hypochondrium. Compared with open surgery, this approach does not require mobilization of the liver (by intersection of the sickle ligament); the rest of the surgical steps are identical. The mini-laparotomy approach combines the advantages of open surgery (the possibility of using higher magnification, enhanced hermeticity of anastomosis and reduced operation times) and laparoscopic techniques (low invasiveness, minimal adhesions in the abdominal cavity and excellent cosmetic results) [Razumovsky A.Yu., Degtyareva A.V., Kulikova N.V., Ratnikov S.A. Advantages of mini-access for Kasai surgery in children with biliary atresia. Surgery. Journal them. N.I. Pirogov. 2019; (3): 48-59. https://doi.org/10.17116/hirurgia201903148] (this information was added to the Materials and Methods section).
- Comment: which type of anti-reflux valve was used and in all patients or only in patients with open surgery?
Response: Construction of the anti-reflux valve was performed only in the patients with open surgery. The procedure involved circular deserosation of a 1 cm length of the intestine at the junction with the Roux loop; the valve was constructed by intussusception with sero-serosal nodal sutures (the information was added to the Materials and Methods section).
Subsequently, the same approach was used for the treatment of recurrent cholangitides that did not respond to conservative therapies. The construction of anti-reflux valve at the Roux loop-intestine junction was aimed at reducing contamination of the portal area with intestinal flora. The procedure involved circular deserosation of a 1 cm length of the intestine; the valve was constructed by intussusception with sero-serosal nodal sutures. (the information was added to the Materials and Methods section).
- Comment: transaminases are not good biological marker of cholestasis.
Response: We fully agree with this comment. Blood levels of transaminases were used by us as auxiliary indicators of normalization of metabolic and synthetic functions of the liver against the background of a decrease in bilirubin levels.
- Comment: some results are not reliable: nobody all around the world has 89.5% of good post-operative bile drainage after kasai procedure. Color of stool and "decreased bilirubin level" are somewhat elusive and subjective parameters. A certain degree of stool staining after surgery should not be mistaken for good post-operative bile drainage,
Response: This is a very serious point. To resolve the misunderstanding, section 3.1 ‘Restoration of the liver function’ was substantively revised. The title of Table 1 now reads as ‘Blood test indicators before the surgery and on day 10 p/o for the patients with the coloration of stools in the early post-operative period (n = 128)’. We fully agree that the appearance of stained stools does not indicate the quality of bile drainage, although it confirms the primary success of the operation in the form of establishing a physical connection of the biliary tree with the intestine. We also fully agree that a clinically relevant assessment of the outcome is possible only in the retrospect at a later date. Nevertheless, the primary improvement must be recorded, since it is an absolutely necessary, although not sufficient, criterion for the success of the operation and the only available method for the monitoring of complications in the early postoperative period.
- Comment: Authors found no difference in the incidence of cholangitis in patients with and without good post-operative bile drainage. This is in conflict with the experience reported in the literature by all the Authors according to whom cholangitis occurs almost exclusively in patients with good biliary drainage. Besides, although cholangitis itself is one of the main causes of the cessation of biliary drainage and failure of the Kasai procedure, Authors do not mention how they try to prevent its onset
Response: In the studied cohort, cholangitis was actually diagnosed in the patients with unsuccessful surgery within 1 year after the operation. In order to prevent cholangitis post-operatively, prophylaxis with trimethoprim-sulfamethoxazole was administered on a long-term basis (this information was added to the Materials and Methods section).
- Comment: the data presented in the tables are not clear
Response: The tables were reformatted and all efforts were made to improve the layout.
- Comment: since one of the still most debated issues is the use of laparoscopic technique compared to open technique, in the analysis of the results Authors do not analyze the results of the different techniques.
Response: Regrettably not: reliable stratification by the type of operation would require bigger samples since the heterogeneity by other criteria is too great.
- Comment: about portal hypertension: do the authors perform routine endoscopic evaluation? If no, which are the indications for scoping?
Response: Endoscopic examination was performed routinely once in 1–1.5 yrs or upon appearance/aggravation of hematological (thrombocytopenia and / or pancytopenia) and sonographic signs (enlarged spleen, recanalized umbilical vein, decreased rates of portal blood flow, increased rates of splenic venous blood flow, splenomegaly, ascites) of portal hypertension (this information was added to the Materials and Methods section).
The manuscript was proofread and the invalid sources were replaced with relevant analogs.
A .docx copy of the revised manuscript with highlighted changes is enclosed.
Best regards,
Dr Denis V Rebrikov
Corresponding Author
Reviewer 2 Report
This paper evaluated the functioning of the liver, the survival rate as a function of age during surgery, and the age-related structure of complications in the follow-up of Kasai surgery. The findings highlighted the crucial value of Kasai surgery surgical correction of BA within the first 60 days of existence and subsequent dynamic patient follow-up for the intent of early diagnosis and prompt correction of possible complications. The manuscript provides substantial evidence for its conclusion and may serve as an interesting read to the journal’s readers. Major issues are missing information on conceptual advance over previously published work, insufficient methodological details, and casual referencing. It is important to revise the manuscript with the following comments:
Major comments
- There are reasonable numbers of publications available on Kasai portoenterostomy for biliary atresia. The authors should expand the introduction and discussion sections to indicate what extra information this manuscript provides to advance in understanding and influence thinking in the field.
- It is important and useful to clearly indicate the inclusion criteria for the subject recruitment.
- The authors should avoid casually citing the references. Several statements in the manuscript need to be backed-up with references, few examples are: Page1- line 31 and 39; Page 9- line 312 and 314; Page 10- line 325
- The discussion section would benefit from expanding the clinical relevance, and referring the reader to possible adverse effects and limitations of the Kasai portoenterostomy approach.
Minor comments
- Page 2, line 51: This sentence ‘The BA diagnosis was confirmed by a standard complex examination for all cases’ should read as ‘The BA diagnosis was confirmed by a standard complex examinations for all cases’
- Page 13, line 137: In the sentence ‘The age- related dynamics of the liver function indicators evaluated with the exclusion of liver transplant recipients is given in Table 2’, consider changing ‘is’ to ‘are’.
- Page 5, Figure 1: Correct the typo ‘Portal hypertansion’ in x-axis label in the bar plot. Also, it is important to display error bars on the bar plot.
- Page 6, line 205: In this sentence ‘Two patients, 1.5 and 2.5 year old..’, consider changing ‘year’ to ‘years’
- Several places in the manuscript indicated ‘Reference source not found’: Page 1- line 30; Page 9- lines 268, 270, 280, 297, 299, 300, 301 and 302; Page 10- lines 317, 319, 326 and 334. Authors should fix these errors and carefully check the reference numbering.
- Figures 2, 3, and 4: Displaying the survived patients out of the total number of recruited subjects on survival plots would make it an easy read.
- Page 10; line 352: In this sentence, ‘The risks of developing cholangitides are high during the first 3 years of life and decrease with the patient's age.’, consider changing ‘decrease’ to ‘decreases’.
Author Response
Dear Reviewer,
Thank you for the thorough consideration and valuable comments on the manuscript.
The manuscript was proofread, the grammar was checked and corrected as suggested in the Review and the invalid sources were replaced with relevant analogs.
Inclusion and exclusion criteria for the recruitment were added to the Materials and Methods section, which was substantively expanded.
A .docx copy of the revised manuscript with highlighted changes is enclosed.
Best regards,
Dr Denis V Rebrikov
Corresponding Author
Reviewer 3 Report
Reference sources must be valid!
Author Response
Dear Reviewer,
Thank you for being so positive.
The manuscript was proofread and the invalid sources were replaced with relevant analogs.
Best regards,
Dr Denis V Rebrikov
Corresponding Author
Round 2
Reviewer 2 Report
The current revision and author’s response to previous comments are acceptable. The authors are advised to carefully proofread the manuscript.
Author Response
Thanks a lot for the comments. The manuscript has been carefully proofread